# Stakeholders perspective of, and experience with contact tracing for COVID-19 in Ghana: A qualitative study among contact tracers, supervisors, and contacts

**Nashira Asiimwe[1,2], Philip Teg-Nefaah Tabong[3]\*, Stanley Amogu Iro[4], Charles Lwanga Noora[2], Kwabena Opoku-Mensah[3], Emmanuel Asampong[3]**

**1** Pharmaceautical Society of Uganda, Kyambogo, Kampala, Uganda, **2** Department of Epidemiology, School of Public Health, College of Health Sciences, University of Ghana, Legon, Accra, Ghana, **3** Department of Social and Behavioural Sciences, School of Public Health, College of Health Sciences, University of Ghana, Legon, Accra, Ghana, **4** West African Centre for Cell Biology of Infectious Pathogens (WACCBIP), University of Ghana, Legon, Accra, Ghana

\* philgh2001@yahoo.com

## Abstract

### Background

Ghana confirmed the first two cases of Severe Acute Respiratory Syndrome Coronavirus 2 (SARS CoV-2) infection on 12[th] March 2020. Following this, the government introduced routine and enhanced contact tracing to identify, quarantine, and test contacts for COVID-19. This study, therefore, intends to document the experiences of contact tracers, their supervisors, during COVID-19 containment in Ghana.

### Methods

Purposive sampling was used to select twenty-seven (27) participants; sixteen contact tracers, six supervisors, and five contacts of COVID-19 cases for an in-depth interview using a topic guide. These interviews were conducted on a phone or face-to-face basis whilst maintaining physical distancing protocol. All these were recorded and transcribed verbatim. Then, QSR NVivo 12 was used to analyse the data thematically.

### Results

Contact tracers were selected based on their professional background and surveillance experience with other infectious diseases. They were trained before the first confirmed cases of COVID-19 in the country and before deployment. Deployment of contact tracers was in pairs to monitor contacts daily through physical visits or over the phone. Their activities included educating contacts about the condition, filling the symptoms diary, and providing psychological support. Contacts for COVID-19 were identified through case investigation, and their monitoring is done once a day despite the twice-daily requirement. Wherever a case was confirmed, enhanced contact tracing within a 2km radius was done. Furthermore, it was reported that some contacts were not adhering to the self-quarantine. In

**Data Availability Statement:** This study was conducted at the beginning of the COVID-19

pandemic and we have provided detailed socio-demographic and professional backgrounds of all the participants in the manuscript. Providing the individual transcripts will breach the confidentiality and anonymity requirement during ethical approval. As a result, we do not have permission from the participants to share the raw data. All interested researchers/readers/persons who meet the criteria for access to confidential data can access the data set from the Ethics Committee for Humanities of the University of Ghana on ech@ug.edu.gh or the corresponding author via this email address: philgh2001@yahoo.com or ptabong@ug.edu.gh.

**Funding:** No funding was received for this study.

**Competing interests:** No authors have competing interest.

addition to this, other challenges included; unstable provision of PPEs and remuneration, refusal of some contact to test, delays in receiving test results, and poor coordination of the whole process.

## Conclusions

The study concludes that contact tracing was generally perceived to be helpful in COVID-19 containment in Ghana. However, adhering to self-quarantine protocol had many challenges for both contact tracers and the contacts. Improving coordination and quick release of test results to contacts is necessary for COVID-19 containment. Lastly, the supply of Personal Protection Equipment and motivation needs to be addressed to help position the country well for effective contact tracing.

## Introduction

A novel human coronavirus, Severe Acute Respiratory Syndrome Coronavirus 2 (SARS-CoV-2), was identified in China in December 2019 which cause a disease known as COVID-19 [1]. COVID-19 is due to a new virus to which no one has immunity. The incubation period has been reported to range from 2 days to 21 days, but with many people reporting symptoms within 14 days [2]. A study from China suggests a severity profile of about 80% mild cases (including subclinical), 13–15% moderate to severe cases requiring oxygen supplementation and hospitalisation and up to 5% critically ill, requiring Intensive Care Unit (ICU) support [3, 4]. A study from China Centre for Disease Control (CDC) showed the majority of patients (80.9%) were considered asymptomatic or presented with mild pneumonia but shed tremendous amounts of the virus at the early phase of infection, which posed an enormous challenge for containing the spread of COVID-19 [5].

On 11th March 2020, the World Health Organization (WHO) declared COVID-19 a global pandemic due to the deep concern both by the alarming levels of COVID-19's spread and severity, and slow global response [6]. This was evident globally because as of 27th June 2020, COVID-19 had affected 215 countries and territories, with a total of 10,069,482 confirmed COVID-19 cases and 500,579 deaths [7]. In Africa, the disease has affected 52 countries with a total of 360,427 cumulative cases, 9,291 deaths and 173,267 recoveries [8].

Transmission of infectious disease relies on three conditions: sources of infection, routes of transmission, and susceptible hosts. According to the National Health Commission of China, in the sixth version of the guidelines for diagnosis and treatments for COVID-19, SARS-CoV-2 was transmitted through respiratory aspirates, droplets, faeces and contacts, and aerosols transmission is highly possible [5]. COVID-19's mean reproduction number $R_0$ is estimated around 2.24–3.58 [9], with a hospitalisation rate of 4.6 per 100,000 population which is higher among adults aged $\geq$65 years (13.8%) [10].

Ghana confirmed the first two cases of COVID-19 on 12th March 2020. Since then, the case incidence has risen suddenly. As of 27th June 2020, a total of 16,431 cases had been confirmed with 103 deaths with at least a case confirmed in all the sixteen regions in the country. In response to the COVID-19 pandemic, the government rolled out Ghana's Emergency Preparedness and Response Plan (EPRP) with the objective to "enhance surveillance system and build response capacity to detect, contain, delay and respond to a COVID-19 outbreak in Ghana". It was an adaptation from the strategic Preparedness and Response Plan for COVID-19 by WHO for countries around the world with the overall objectives to (a) slow and stop

transmission, prevent outbreaks, and delay spread; (b) provide optimised care for all patients, especially the seriously ill; and (c) minimise the impact of the pandemic on health systems, social services and economic activities.

One strategy adopted in Ghana for the containment of the COVID-19 is contact tracing. Contacting tracing is a monitoring process to identify people who have come into contact with cases and instituting measures to reduce their interaction with other people as well as screen them for possible infection. This process involves three steps; contact identification, listing, and follow-up of contacts. The first step is contact identification which involves asking the infected person about his/her activities and the roles of the people around him/her from two days before the onset of symptoms. This can be done either on a face-to-face basis or remotely through a phone call. A contact can be anyone who has been in close proximity (direct physical contact or having shared an enclosed space) with an infected person: health care providers' family members, friends, or work colleagues. The second step involves the contact listing. In this step, all persons considered to have come into contact with an infected person should be listed as contacts. In this regard, efforts are made to identify each contact listed and to inform them about the exposure status and the need for contact tracing. All contacts are advised to quarantine whilst waiting for their test results. If the test results are positive, isolation is required either at home, in the hospital, or at a designated centre. The final stage is a follow-up for all contacts. Regular follow-ups are conducted with all the listed closed contacts to monitor for symptoms and symptoms, and test for evidence of infection [11].

When contact tracing is effectively implemented, it can lead to a better containment of the infection. As part of the ongoing efforts for Ghana to contain COVID-19, we explored views of COVID-19 contacts, frontline contact tracers, and their supervisors on the contact systems implemented in Ghana to inform decision-making.

## Materials and methods

### Ethics statement

The protocol for the study was reviewed and approved by the Ethics Committee for Humanities, University of Ghana, Legon. Written consent was obtained from all participants and data collected anonymized.

### Study design

We adopted a qualitative approach to research. Specifically, we used phenomenology and narrative study designs. Listed COVID-19 contacts who were either asked to self-quarantine or quarantine at designated places, shared their lived experiences with the process as required in the phenomenology approach to research [12]. However, contact tracers and supervisors shared their experiences and knowledge about how the process is being conducted using the narrative approach [13]. In conducting the study, we adhered to the consolidated criteria for reporting qualitative research (COREQ) [14], and acceptable practice in fieldwork, analysis, and interpretation [15].

### The theoretical framework for the study

We adopted Lipsky's Street-Level Bureaucrats Theory that highlights the role that frontline health care providers play in achieving policy objectives. The contact tracing was adopted as one of the strategies to help in the containment of the COVID-19 in Ghana. Lipsky's street bureaucrats theory argue that frontline health workers do not always implement policies developed by their superiors to the level of fidelity required by the policy [16]. The implementers of

a given policy use their discretion to circumvent gaps between policy requirements and resources available. This gap creates a conflict that needs to be resolved. Lipsky is of the view that the implementers adopt what is called "coping mechanisms" to fill the available gaps. The bureaucrats include; health workers, teachers, public lawyers, social workers, judges, police officers, and other public service employees who provide government services, enforce the law, and distribute public benefits to citizens directly [16].

Even though this theory has been widely adopted in health system research, critics of the street-level bureaucracy theory argues that Lipsky places a lot of discretional powers in the hands of public service workers. However, these same critics acknowledge that workers play an important role in policy decisions and implementation strategies and do not merely implement decisions made by their superiors [17]. This theory was useful in this study as it provided guidance on how the contact tracing is being conducted. Furthermore, it used to determine the stakeholders to select for this study.

## Study area

The study was conducted in the Greater Accra region of Ghana. This region is the national capital of Ghana and the place where the first two index cases were reported. The national facility for managing people with COVID-19 is also located in this region. The Greater Accra region has a high population density which has the potential to increase the transmission of infectious diseases [18, 19].

## Study population

The study included both males and females within the legal age of 18 years and above for informed consent in Ghana [20]. These included health workers conducting the contact tracing for COVID-19, their supervisors, and selected individuals identified as contacts to confirmed COVID-19 cases.

## Selection of study participants

A purposive sampling technique was used to select contact tracers for this study. The lead supervisor provided the list of contact tracers. These contact tracers were subsequently reached via phone and informed about the study. Those who agreed to take part were interviewed at a prearranged time physically in a one-on-one interview.

We used contact tracers to recruit contacts who were under surveillance. The contact tracers first informed the contacts about the study and requested permission—the phone numbers of those who agreed to participate in the study were then given to the researchers. These contacts were reached on the phone, requested for their consent about recording the interview and those who agreed were interviewed accordingly.

For the supervisors, we used snowball sampling starting from the lead supervisor. Each gave us two names and phone numbers of his/her colleagues after consulting them. The researcher called and set up the interview appointment. In all 27 people were selected for the interview; six supervisors at the national level, 16 contact tracers, and five contacts. Table 1 provides a summary of the various study participants.

## Data collection tool

We prepared an interview guide before conducting semi-structured interviews to ensure that the same basic lines of inquiry were pursued. Three different interview guides were developed; one each for supervisors, contact tracers, and contacts (see S1–S3 Files). The interview guide

**Table 1. Category of study participants.**

| Category of interviewee | Number interviewed |
|---|---|
| Supervisors | 6 |
| Contact tracers | 16 |
| COVID-19 contacts | 5 |
| Total | 27 |

for supervisors covered areas such as contacts identification, selection of contact tracers, and the strategies employed to get in touch with contacts. The guide also contained questions on contact tracer training, the supply of contact tracer's logistics, and evaluation of the contact training in Ghana as a whole. The topic guide for the contact tracer elicited information on the training content, how they perform contact tracing, actual activities when contacts are visited, and their appraisal of the process and challenges they face. For contacts, the topic guide covered the experience and the type of services they receive from the contact tracers. All the interview guides were in English.

## Data collection procedure

The interviews were all conducted in English with some conducted at the office of selected participants. Physical distance was maintained throughout the face-to-face interviews in their offices. Besides, both the interviewee and interviewer wore a face mask. In some select cases, the interviews were conducted on the phone. Participants were encouraged to dialogue about their experiences with COVID-19 contact tracing. We provided inputs to guide the conversation. It took between 30 to 70 minutes to complete an interview session. All interviews were conducted by two of the authors (NA and ISA) who are postgraduate students with training in qualitative research methodology. After the interview, a summary of the interaction highlighting key points was done to validate the data as a form of member checking [21, 22]. Daily interviews were shared with other authors to review and provide feedback on the process. This iterative approach strengthened the data elicitation process. Interviews continued until data saturation was achieved [23]. We also wrote comprehensive fieldnotes on a daily basis. These field notes were incorporated into the data set. There were a total of six non-respondents (five contact tracers, one supervisor), one of whom didn't want to be recorded and the rest declined due to their busy schedules. Nonetheless, these were replaced until data saturation was reached [15]. The data were collected between 6th and 26th April 2020.

## Data analysis

The interviews were transcribed verbatim and validated by the participants [21]. We read through the transcripts several times to familiarise ourselves with the data. A hybrid inductive and deductive framework was used to develop a codebook. Conceptual dimensions of the interview guides guided the preliminary development of the codebook. This was then revised to include the emerging themes from the data. This codebook was discussed and accepted by all authors. The codebook served as a guide for the thematic analysis [13]. The transcripts were imported into NVivo 12 for coding led by PTNT. We created notes from the codebook. Each transcript was opened in the software, coded, and reviewed. The socio-demographic characteristics (sex, age, and profession) of the respondents were captured as attributes in NVivo, and appropriate values assigned using a classification sheet. This allowed us to make a comparison of data across various respondents and socio-demographic groups. Initially, coding was done into free nodes. However, as the coding progressed, the relationship between nodes became

more apparent, and the free nodes were transformed into tree nodes. In working our way through the data thematically, we used the Nvivo function "memo" to capture unique opinions, poignant experiences, and relevant illustrative quotes. The coded data including the audit trail was shared with other authors who reviewed and made inputs. These steps were taken to improve the trustworthiness during the thematic analysis [24]. Coded sections were regrouped into relevant categories and themes for presenting the results. Direct quotations were used, where appropriate, to support the themes. Data triangulation was used to compare and contrast responses across the three different groups of participants; contacts, contact tracers, and supervisors. Table 2 shows the main themes and subthemes that emerged from the data.

## Results

### Socio-demographic characteristics of participants

All the supervisors and contact tracers had attained tertiary education. Except for one contact, the remaining had completed tertiary education. The participants adhere to two main religious affiliations, Christianity, and Islam. All the supervisors and contact tracers had training in public health. Only five of the 27 participants were female (Table 3).

### Identification of contacts for COVID-19

Contact tracers and supervisors defined a contact as anybody who shares any form of close physical space with a confirmed COVID-19 case/person. The person could be a friend, colleague worker, bordered the same transport, sleeping in the same house, house-keeper of a confirmed case, or provided health care to the cases. The proximity of the confirmed case to the contact is used to determine the risk profile of the person. A supervisor shared his views on who a contact is as follows:

> "A contact is someone who has physically interacted with a person who has tested positive. We have two contacts–close-contact and distant contacts. We also use the WHO approach. When you identify someone as a suspected case and is confirmed to have COVID-19, a health worker goes to the person to do the listing. We help the person think about all his/her activities two days before the test was confirmed. This is done day by day" (S003)

**Table 2. Main themes and subthemes from the data.**

| Main themes | Subthemes |
|---|---|
| Identification of COVID-19 contacts | • Definition of contact |
|  | • Types of contacts |
|  | • Process of identification |
| Selection and training of contact tracers | • Selection |
|  | • Training |
|  | • Content of training |
|  | • Relevance of training |
|  | • Deployment of contact tracers |
| Contact tracing strategies | • Contact tracing |
|  | • Data collection |
|  | • Education provided to contacts |
| Compliance with quarantine process | • Motivators |
|  | • Challenges |

**Table 3. Socio-demographic characteristics of participants.**

| ID | Sex | Age | Professional Qualification | Marital Status | Religion |
|---|---|---|---|---|---|
| **Supervisors** | | | | | |
| SI001 | Male | 42 | Medical Doctor/Field Epidemiologist | Married | Christianity |
| SI002 | Male | 33 | Biomedical Scientist/Field Epidemiologist | Single | Christianity |
| SI003 | Male | 35 | Epidemiologist | Married | Christianity |
| SI004 | Female | 33 | Public Health Specialist | Married | Christianity |
| SI005 | Female | 35 | Medical Doctor | Single | Christianity |
| SI006 | Female | 38 | Medical Doctor | Single | Christianity |
| **Contact Tracers** | | | | | |
| CT001 | Male | 35 | Disease Control Officer | Married | Christianity |
| CT002 | Male | 35 | Surveillance Officer | Married | Christianity |
| CT004 | Female | 38 | Nurse | Married | Christianity |
| CT003 | Male | 36 | Surveillance Officer | Single | Islam |
| CT005 | Male | 42 | Public Health Nurse | Married | Christianity |
| CT006 | Male | 24 | Nurse | Single | Islam |
| CT007 | Male | 32 | Surveillance Officer | Married | Islam |
| CT008 | Male | 32 | Disease Control Officer | Married | Islam |
| CT009 | Male | 30 | Laboratory Scientist | Married | Islam |
| CT010 | Male | 50 | Senior Community Health Officer | Married | Islam |
| CT011 | Male | 33 | Disease Control Officer | Married | Islam |
| CT012 | Male | 32 | Veterinary Laboratory Technician | Married | Islam |
| CT013 | Male | 30 | Teacher | Single | Christianity |
| CT014 | Male | 33 | Public Health Nurse | Married | Christianity |
| CI015 | Male | 34 | Medical Doctor | Married | Christianity |
| CT016 | Male | 31 | Public Health Specialist | Single | Christianity |
| **Contacts** | | | | | |
| C001 | Male | 51 | Financial Organization Personnel | Divorced | Christianity |
| C002 | Male | 65 | Retiree | Divorced | Christianity |
| C003 | Female | 50 | Financial Organization Personnel | Married | Christianity |
| C004 | Male | 28 | Cleaner | Single | Christianity |
| C005 | Male | 60 | Management Specialist | Married | Islam |

Contacts for COVID-19 are identified through case investigation. This involved taking the history of all confirmed cases, his/her movements, and activities. It gives a clear indication of the people the case would have encountered as explained by one supervisor:

*"We use case investigation forms-travel history, attending a health facility and treated for any illness prior to being confirmed"* (S002)

Wherever a case is confirmed, all people who live in a two kilometres radius of the residence of the confirmed case are screened. This is done as a form of enhanced contact tracing. One supervisor's explanation is as follows:

*"We introduced a strategy in our enhanced contact tracing and testing. Because some confirmed cases are asymptomatic, we take a certain radius of all people within the vicinity of the confirmed case. This is done with the belief that the confirmed case may have closely related to people within this radius"* (S001)

A line list of all the contacts is done, and initial samples are taken for the test. After that, daily monitoring is done. This monitoring is done twice daily—morning and evening. During the contact monitoring, the temperature of the person, other socio-demographic and illness history are taken. After fourteen days, the samples are taken for a repeat test to be conducted. A supervisor shared his experience on how monitoring is done:

*"The monitoring is supposed to be done twice a day, but in many instances, the monitoring is done once a day. During the contact, the contact tracer fill the case-based form and checking of temperature"* (S005)

## Selection, training, and deployment of contact tracers

The study showed that contact tracers included public health physicians, field epidemiologists, disease control officers, nurses, and volunteers. Contact tracers were selected based on their professional background and those involved in already existing surveillance chains for other endemic infectious diseases. At the community level, there are community health nurses as contact tracers. At the district level, there are disease control officers with epidemiologists and clinicians at the regional level. These cadres of health workers already have prior training on filling case-based forms. All the contact tracers were health workers with some level of training on infectious diseases as confirmed by some supervisors in the following statements:

*"We look for a cadre of staff who qualify to be contact tracer. We look for people with basic training in public health, understand the health system. Each district identify their contact tracers"* (S004)

*"All our contact tracers are health workers who have formal training"* (S003)

Contact tracers in this study corroborated that they had received training before they were deployed. A contact tracer shared his view about the training as follows:

*"We were given some basic training on the contact tracing. We were taken through the case-based forms and the dos and don'ts in contact tracing. We were informed about how to protect ourselves from infection, not eat or drink anything from a contact"* (CT003)

The content of the training includes signs and symptoms of COVID-19 and completing the symptoms diary and daily monitoring of charts:

*"We go through the symptom's dairy and daily monitoring charts with them. These charts come in two forms, paper-based and electronic"* (S002)

The training also covered the definition of contact, how to do contact listing and tracing. The tracers were trained on the use of the infra-red thermometers, data collection and use of personal protective equipment (PPE). The contact tracers were also trained on how to use the Surveillance Outbreak Response Management and Analysis System (SORMAS) and ARC GIS software. A supervisor and contact tracer shared their experience on the content of the training:

*"We trained all the contact tracers. They were trained before first confirmed cases in the country. . .after the detection of the first two cases, another training was done before they were deplored"* (S001).

*"We received training twice, the first one occurred before the first cases and later after the first two cases were confirmed. We were trained on the use of the SORMAS software"* (CT001)

Some contact tracers in an interview indicated they had had prior experience in conducting contact tracing. Some were already engaged in contact tracing for diseases such as tuberculosis, Ebola, and other infectious diseases. One contact tracer shared his experience:

*"I have been involved in contact tracing. During the Ebola outbreak in Liberia, I was involved in contact tracing. So, I have some experience in contact tracing"* (CT0015)

All contact tracers were unanimous that the training was very useful in empowering them to conduct contact tracing without fear of being infected.

*"The training actually was very useful. I have learned how to use the SORMAS and GIS software and this would help me in future"* (CT0012)

Deployment of contact tracers was in pairs in order for them to support each other while in the field in terms of workload and to remind each other about safety measures while on duty. This was emphasized by a contact tracer:

*"We move in pairs. That is important as you can be reminded by a colleague about some safety measures"* (CT008)

## Strategies used in contact tracing

Participants in this study indicated that contact tracers are assigned several contacts after listing. Two main strategies are used in the monitoring of the contacts; daily visits to the contact and remote monitoring for people that preferred this model and had the appropriate knowledge and gadgets like a calibrated thermometer. Following an initial phone call by a supervisor, the details of contacts were passed on to a contact tracer who would make follow up calls and visits to the contact during their quarantine period to check how he/she is doing. One supervisor threw more light on this process:

*"Contact tracers are assigned a number of contacts to monitor them daily. The contact tracers visit the contact to monitor them. . . for people who are health workers and can conduct self-monitoring, we call them on the phone. For these people, the contact tracing is done remotely with an occasional visit to the house"* (S003)

During the phone calls, the symptom diary is completed by the contact tracer with the information provided by the contact on the phone to facilitate the process, as one supervisor explained:

*"For contacts who are health workers who have a personal thermometer, we provide them with the symptoms diary to do self-monitoring at home. The contact tracer calls the contact daily to conduct the monitoring. However, occasional physical contact tracing is done for such people as well"* (S004).

For contacts who receive a daily visit by their contact tracers, the symptoms diary is completed at the residence of the contact. As a safety measure, contact tracers were cautioned against touching any physical object at the house/home of a contact. The contact is called to

inform the person that the contact tracer will be coming to the house. Once they reach the house, the person is supposed to come and open the gate and doors. Contact tracers are also provided with PPEs (such as face masks and gloves), case-based forms, or the tablet where the SORMAS has been installed. One supervisor and a contact tracer stated as follows:

> *"For our contact tracers, we give them masks, hand sanitisers. . . they wear the mask and also maintain a social distance with the contact. . ."* (S005)

> "We are not supposed to touch anything in the house of the contact. We call them to come and open the gate and move backward before you entered. Once you enter, you maintain physical distance, ask about the person's health, and complete the symptoms diary. We also educate them on the condition and the need to self-quarantine" (CT006)

Fig 1 shows a designed framework of the description of how the contact tracing has been implemented by supervisors

Some contact tracers interviewed indicated that some contacts were not adhering to the self-quarantine. This, in the opinion of a contact tracer, could negatively affect containment efforts by Ghana.

> *"There are some contacts that I have been monitoring who are not observing self-quarantine. You go to their house, and they have moved out. When you call to inform them that you are coming, they rush to the house. So, for some of them, I get to the house before making the call as a strategy to monitor if they were observing self-quarantine"* (CT009).

Some contacts indicated it was difficult to adhere to self-quarantine especially where one had to go out to get food and other necessities of daily living, hence the current advocacy for quarantine at a designated place where they can be provided with food and other items is good. A contact shared his experience as follows:

> *"Man has to survive. . .go get food from outside, and friends may still come to you. So, it is really difficult to stay without interacting with anybody for the 14 days"* (C001).

Aside from these challenges, participants were unanimous that the contact tracing process is a useful containment strategy. In addition, it also provided the opportunity for contacts to receive psychological support.

> *"The contact tracing is very good. We get counselling, education, and support. Overall, I think it is great to see people devote their lives and risk to daily come to you to ask about your health"* (C004)

### Motivation for self-quarantine

The motivators for self-quarantine identified in the study include the desire to protect loved ones, the aspiration to reduce disease spread, and lockdowns.

> *"I self-quarantine because I did not want to infect my family and I also saw the need to help contain the disease in the country"* (C001)

COVID-19 related stigma also served as a motivator for self-quarantine. One stakeholder shared his views as follows:

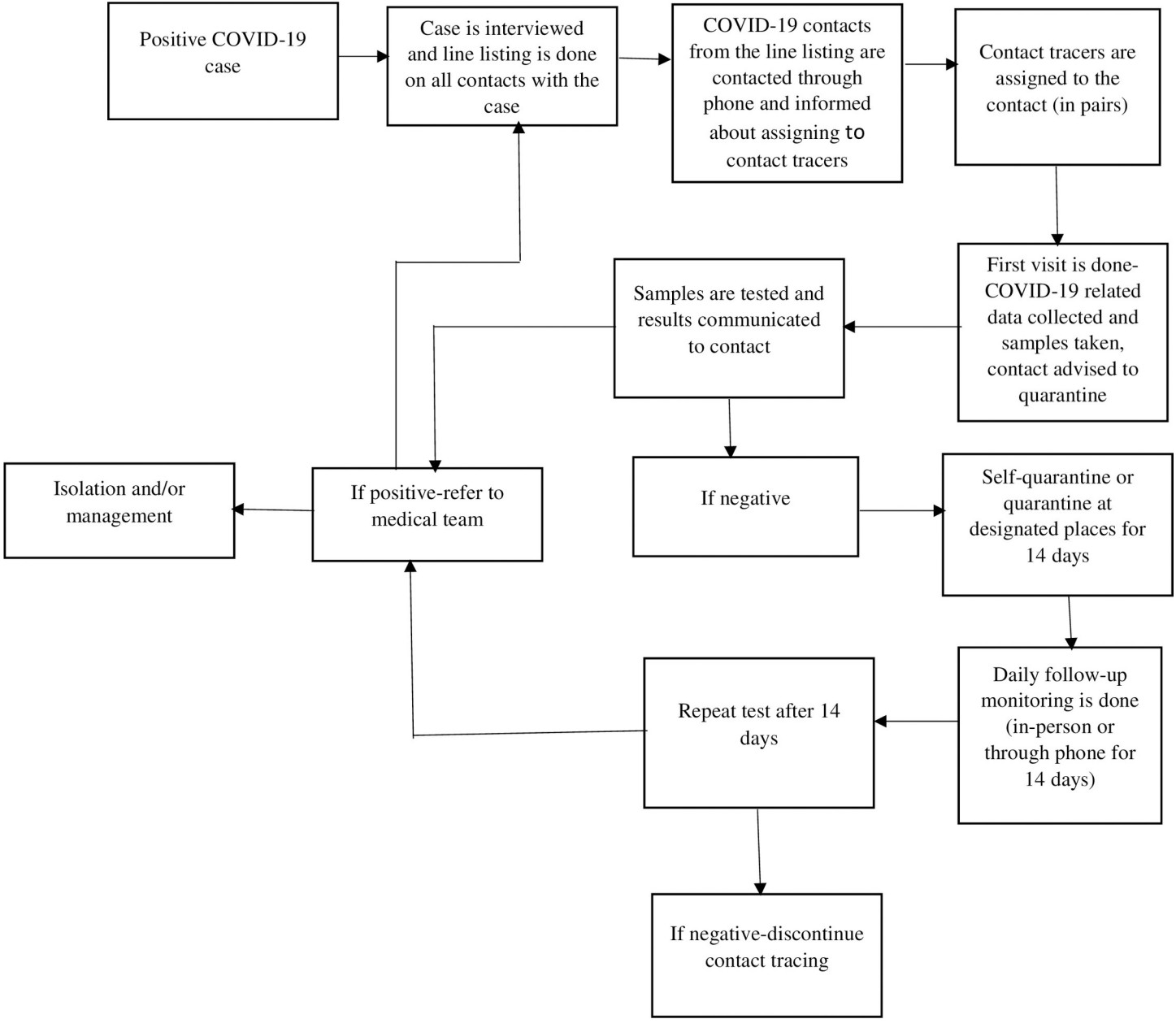

**Fig 1. Framework of contact tracing for COVID.**

*"You know it is a new disease, and people who are infected are stigmatized. So people self-quarantined to stay away from their community members so they would not be identified or frequently seen in public after having been tested positive"* (CT005)

Education and counselling provided to contacts also served as motivators for compliance with the directive to self-quarantine. A contact shared her motivation as follow:

*"The counselling and education I received from the contact tracer encouraged me to restrict my movements and complied with self-quarantine directives"* (C003)

The initial contacts in Ghana were people who belong to higher socio-economic status. As a result they had the resources to self-quarantine as one of them stated:

*"I have my own house which is big enough that enabled me to quarantine in one area whilst the remaining members of the family restricted themselves to the other part of the house"* (C002)

Another contact revealed that her employers assured her of receiving all benefits due to her whilst in quarantine. Hence, it was easy to comply as indicated:

*"My employers sent a well-wishes message to me. They told me that all my salary and allowances would be paid even though I have not been coming to work because of the situation I find myself"* (C003)

At the time of the study, the death toll among countries in Europe, America, and Asia that were first affected by the pandemic was so high, and many had anticipated it to be worse in Africa. This was mentioned as a reason for compliance with the directive to self-quarantine.

*"You see how people are dying in developed countries and the situation is likely to be worse in Africa. So I have to self-quarantine to prevent my family and close relatives from getting infected in case I am tested positive"* (C004)

## Contacts experience with the health workers during contact tracing

Contacts revealed that they were initially afraid and felt uncomfortable when they were told they had to self-quarantine and screen for COVID-19. This directive, according to participants, was informed by the high level of COVID-19 related mortality that had been reported across the world. They indicated that the most troubling part was having to stay in your room and not interact with your family. This, according to contacts in this study, saddened their hearts and left them thinking for several days.

*"I was initially very anxious when I was told I was a contact and had to be screened for the condition. It is not an easy experience, especially the way people were dying from the condition outside the country. I had to also live in my room alone, and food is virtually left at the door for me"* (C002)

COVID-19 contacts revealed they were delighted with the work of contact tracers. To the study participants, the contact tracers were very professional and provided them with the necessary health education and support. All contact tracers also wore a face mask. A contact shared his experience as follows:

*"They [contact tracers] call to explain to me what they will be doing. . . they came wearing masks and maintained some distance from me. . .in all, I will say they were very professional"* (C002)

In the same way, contact tracers generally were of the view that contacts were very cooperative in their engagements with them:

*"My experience with all the contacts I have traced so far have been cooperative. They have been able to provide me with all the information I required"* (CT0011).

*"It was a man and woman; they try to engage you in conversation. They go through the various symptoms, and the temperature check. . .they were calm and provided some psychological support"* (C004).

Nonetheless, some contacts indicated they did not like the daily visit because of the stigma. As a result, some contacts indicated they had to procure their own thermometer to prevent the daily visit by the contact tracers. One of them stated as follows:

*"They were coming every day, but I complained. I told them not to come because there is a stigma. If you sell something, people will not buy it. So, I bought my own thermometer and asked them to call for me to provide them with the information. You know, people in the community would be wondering why health workers come to your house everyday. . . the stigma is there"* (C005).

Some contacts also refuse to pick the calls of the contact tracers. This was, therefore, in a way, undermining the process. A supervisor shared the following in an interview:

*"Sometimes your contact tracer come and tell you how some contacts have insulted them. There are lots of psychological dimension to it . . .so I encourage the contact tracers not to give up. Some people believe they have low risk so when you list them and want to monitor them, they relocate, or you call them, and they will refuse to pick, and when you go to the house, you don't find them"* (S006).

## Challenges and lessons learned in the contact tracing

Five main challenges were identified; provision of PPEs, remuneration, and refusal of some contact to testing, poor coordination, people not adhering to personal protection etiquettes, and quarantine. Some contact tracers expressed concern about the type and inadequate provision of PPEs for personal use. During the early stages of the epidemic, participants indicated that contact tracers were basically concern with daily monitoring of contacts. However, as the number of cases increased, contact tracers had to take up the responsibility of taking COVID-19 samples. This according to contact tracers, required more advanced PPEs that were unavailable. Two contact tracers shared their views on the provision of PPE as follows:

*"We are only given facemask and hand sanitisers, but now some of us take samples and hence the risk is high. So, we need more advanced PPEs like N95, eye shields and biohazard suits"* (CT0011).

*"We do not have enough PPEs, and you know the work we do is risky. You can get infected because of daily interaction with contacts"* (CT007)

Participants also revealed details of their remuneration. At the beginning of the engagement as contact tracers, they were paid daily. However, as the number of cases increased, more contact tracers were engaged, resulting in delays in payments. Some participants also indicated there was a reduction in the amount they were promised as remuneration even though the workload had increased. The following illustrative quotes support these points.

*"Initially we were receiving a daily allowance, however, as the number of cases increased, more people were recruited and it in a way affected our allowance. Now we do not get the daily allowance"* (CT012).

*"I have been working as a supervisor but, yet to be given the allowance. . . This is negatively affecting the zeal of contact tracers"* (S003)

Another concern raised was enforcing self-quarantine for COVID-19 contacts. Some contacts indicated the living arrangements in their house and/or home made it difficult for them to self-quarantine. However, the absence of a centralised system to keep contacts, compelled them to opt for self-quarantine despite inherent challenges. This, according to some contacts, may lead to exposure of their household members should they eventually test positive. Some contact tracers suggested as follows:

*"Our government has to get a central place to quarantine all contacts. This will be important to ensure people adhere to the quarantine. Some of our living arrangement makes it difficult not to interact with other members of the family"* (CT009)

*"I think if we are able to get quarantine centres for all contacts, it will help to check the people who are contacts from moving around in their community to expose others"* (CT005)

Furthermore, delays in getting test results also emerged as one of the challenges. This, according to respondents, has resulted in some contacts denying the outcome of tests. This has also resulted in a situation where some contacts refuse to undergo the test.

*"The delays in getting the test results are affecting the contact tracing. You take the sample, and it takes several days before you inform the person of the outcome of the test. Because of that, some people will tell you the test results is not for them especially when it is positive"* (CT005)

*"One of my staff shared his experience where the contact insulted them and refused to allow them to take samples and complete the case-based forms. We also have contacts who we are unable to trace once the results are back. Others will not also tell you the possible contact"* (S006)

Lastly, participants indicated a lack of coordination in the entire COVID-19 response and contact tracing. The contact tracing platform and the testing results were managed by two different groups and platforms making coordination difficult. This was cited as one of the reasons for the delays in receiving test results and its associated negative effects on the process. In the view of some supervisors and contact tracers, there was deep-seated politicisation of the response to COVID-19. Transparency was, therefore, a problem that resulted in dealing with issues of truth among both contact tracers and contacts. As the cases spread to other regions in the country, the contact tracing was decentralised; however, resources were still managed from Accra. This negatively affected regional and district response to the COVID-19 and contact tracing. The following illustrates these points:

*"Coordination is a big problem in this entire COVID-19 response. Politicians are at the forefront of the fight. Every Ministry tries to do something without a centralised coordination system. So, there is a lack of harmony in the activities"* (S003)

## Discussion

The study was conducted to document stakeholder's perspectives and experiences with the COVID-19 contact tracing. The findings of the study show that contacts were generally

satisfied with the contact training process, nevertheless, concerns were raised about daily home visits. Although the study revealed that the communities and contacts were willing to support contact tracers, concerns were raised about stigmatisation of contacts. Thus, some contacts felt it was inappropriate for contact tracers to pay daily visits. This, in their view, could accentuate COVID-19 related stigma in the community. This is an important observation in this study. There are media reports of treated COVID-19 cases having challenges of re-integrating into society. This stigma could be reduced by providing health education on COVID-19 to the public without sensationalising the situation. This calls for a review of the information and communication strategies that are being used for COVID-19. Fear appeal and sensationalising the situation have been documented as a precursor for stigmatising people with infectious diseases [25, 26].

Self-quarantine was generally practiced, however, some participants were not adhering to this protocol. Those who complied with this protocol did so because they had the resources and also saw the need to protect their family members and close associates. The education and couselling/psychological support provided by the contact tracers encouraged the contacts to self-quarantine. It must be emphasized that the initial counselling was done by health care providers with little training in counselling. It is therefore important to incorporate professional counselors into the programme to ensure that the psycho-social needs of contacts and their families are met. Such supports are necessary to minimize the effects of infectious diseases on both the infected and affected people, and their communities [27, 28].

Participants also raised issues about delays in getting test results and lack of coordination. The present study has further shown that respondents perceived delays in getting test results as a significant challenge in the community. According to the study participants, this resulted in some contacts refusing to take the COVID-test or when tested the outcome of test results. It is important for the country to improve on the interval between the taking of samples and giving feedback on the results. As observed in this study, some contacts were not practicing self-quarantine and this will therefore pose a risk to other susceptible individuals in their community. For an infectious disease with a high basic reproductive rate like COVID-19, it is essential to ensure that test results are communicated to contacts promptly. Again, the delay was also creating mistrust among various stakeholders. Mistrust among stakeholders is often a recipe for the spread of misinformation and disinformation during public health emergencies [29]. This also negatively affected the frontline workers compliance to policy directives as articulated by Lipsky's Street Bureaucratic model [30].

Despite the contact tracers and supervisors' seeming enthusiasm for the work, they were worried about the delay in payment of remuneration and lack of coordination—an essential requirement for the success of the entire COVID-19 response. Initially, contact tracers were paid daily but as the number of cases increased, more people were recruited from within and outside the health system to support the exercise. This led to the delay in the preparation and payment of their remuneration which affected their morale to work. Moving forward, the COVID-19 contact tracing could be integrated into the Community-based Health Planning and Services (CHPS) concept being implemented in Ghana [31, 32]. Under this, the Community Health Officers (CHOs) who pay daily visits to families in their catchment areas as part of their routine work could be resourced to conduct contact tracing activities. This can reduce the stigma associated with the daily visits by other health workers who initially served as contact tracers and also reduce demand for remuneration.

The concerns raised by contact tracers and supervisors about coordination which was negatively affecting the entire process need to be addressed. As the disease spread outside the national capital (Accra) which was the epicenter, attempts were made to decentralize national efforts at containing the spread of the virus. However, the biggest challenge faced was

coordinating the activities of various ministries and departments and the supply of PPEs. There is therefore the need for government to improve coordination. Also decentralizing the response without providing the districts and regions with the needed resources may undermine containment efforts.

## Limitations

Even though this study provides useful insights into the contact tracing process and experience from various stakeholders in Ghana, its findings should be interpreted in the context of some limitations. One weakness in qualitative studies is the inability to generalize the findings [13]. The study was also conducted within the first 45 days of the COVID-19 outbreak in Ghana, therefore could not capture all the dynamics in the contact tracing processes as the situation evolved.

## Conclusion

The study concludes that the contact tracing was generally perceived to be helpful in COVID-19 containment in Ghana. However, adhering to self-quarantine protocol had many challenges for both contact tracers and the contacts. Improving coordination and quick release of test results to contacts is necessary for COVID-19 containment. Lastly, the provision and supply of Personal Protection Equipments and motivation needs to be addressed to help position the country well for effective contact tracing. Measures should be taken to enforce quarantine among COVID-19 contacts to minimize spread to family, work colleagues and community members.

## Supporting information

**S1 File. Key informant interview guide.**
(DOCX)

**S2 File. Interview guide for contacts.**
(DOCX)

**S3 File. In-depth interview guide for contact tracers.**
(DOCX)

## Acknowledgments

We wish to thank all study participants for their corporation and time.

## Author Contributions

**Conceptualization:** Nashira Asiimwe, Philip Teg-Nefaah Tabong, Kwabena Opoku-Mensah, Emmanuel Asampong.

**Data curation:** Nashira Asiimwe, Stanley Amogu Iro.

**Formal analysis:** Philip Teg-Nefaah Tabong, Kwabena Opoku-Mensah, Emmanuel Asampong.

**Investigation:** Nashira Asiimwe.

**Methodology:** Nashira Asiimwe, Philip Teg-Nefaah Tabong, Stanley Amogu Iro, Charles Lwanga Noora, Kwabena Opoku-Mensah, Emmanuel Asampong.

**Resources:** Nashira Asiimwe, Stanley Amogu Iro, Charles Lwanga Noora.

**Software:** Philip Teg-Nefaah Tabong.

**Supervision:** Philip Teg-Nefaah Tabong, Charles Lwanga Noora, Emmanuel Asampong.

**Writing – original draft:** Philip Teg-Nefaah Tabong.

**Writing – review & editing:** Nashira Asiimwe, Stanley Amogu Iro, Charles Lwanga Noora, Kwabena Opoku-Mensah, Emmanuel Asampong.

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
