## [Decision Letter · Decision Letter 0]

2 Nov 2020

PONE-D-20-20303

Stakeholders perspective of, and experience with contact tracing for COVID-19 in Ghana: A qualitative study among contact tracers, supervisors, and contacts

PLOS ONE

Dear Dr. Tabong,

Thank you for submitting your manuscript to PLOS ONE. After careful consideration, we feel that it has merit but does not fully meet PLOS ONE’s publication criteria as it currently stands. Therefore, we invite you to submit a revised version of the manuscript that addresses the points raised during the review process.

The manuscript has been evaluated by three reviewers, and their comments are available below. The reviewers have raised a number of concerns that need attention. They request additional information on methodological aspects of the study and the interpretation of the results. Could you please revise the manuscript to carefully address the concerns raised?

We look forward to receiving your revised manuscript.

Kind regards,

Dario Ummarino, Ph.D.

Senior Editor

PLOS ONE

Journal Requirements:

2. Please include additional information regarding the interview guides used in the study and ensure that you have provided sufficient details that others could replicate the analyses. For instance, if you developed a questionnaire as part of this study and it is not under a copyright more restrictive than CC-BY, please include a copy, in both the original language and English, as Supporting Information

Reviewers' comments:

Reviewer's Responses to Questions

**Comments to the Author**

1. Is the manuscript technically sound, and do the data support the conclusions?

Reviewer #1: Yes

Reviewer #2: Partly

Reviewer #3: Partly

2. Has the statistical analysis been performed appropriately and rigorously? 

Reviewer #1: N/A

Reviewer #2: N/A

Reviewer #3: N/A

3. Have the authors made all data underlying the findings in their manuscript fully available?

Reviewer #1: No

Reviewer #2: Yes

Reviewer #3: No

4. Is the manuscript presented in an intelligible fashion and written in standard English?

Reviewer #1: Yes

Reviewer #2: Yes

Reviewer #3: Yes

5. Review Comments to the Author

Reviewer #1: This is a very nice piece of research about a very important and urgent topic.

In the rush to contain COVID, policy makers need to know what is actually happening in the ground. This paper contributes to that understanding.

The qualitative approach is appropriate as it provides the additional nuance needed to understand this complex multidimensional problem. The sample size is appropriate for this type of research, and the stakeholders interviewed seem to be a good selection.

The writing is clear, although sometimes repetitive. The article could be made shorter.

The major problem is that the themes arising from the study are not clear. The structure of the paper emphasizes aspects of the process. Is this what you found in the thematic analysis? Please explain how the data analysis led to the structure of reporting in the results section. This might need to be adapted to match the themes.

You might find the guidelines in this paper useful:

Nowell, L. S., Norris, J. M., White, D. E., & Moules, N. J. (2017). Thematic analysis: Striving to meet the trustworthiness criteria. International journal of qualitative methods, 16(1), 1609406917733847.

It would also be interesting to identify what might the motivational factors that contribute to people quarantining according to the different stakeholders. These motivational drives are not made clear but would be very useful to policy makers.

Reviewer #2: This manuscript presents important experiences from people involved in contact tracing activities, from supervisors to contacts of confirmed COVID-19 cases. The topic is interesting given the unprecedented crisis the world is experiencing due the outbreak of a novel virus. It is important to share the experience of African countries as their 'community' approach to contact tracing has been different from other countries (maybe even more effective). There are very interesting points discussed in this paper but there are also some parts that could be improved by trying to describe in more detail the nuances of stakeholders' experiences, particularly those of the contact tracers and supervisors. For instance, was the training useful, if not, why? Did they follow all the guidance for contact tracing? If not, why not? Is there any data that can bring more insights into supervisors and contact tracers' experiences?

Also, there are some important points that were not fully discussed in the discussion section and could be addressed better.

Make sure to ask someone to proofread the manuscript before sending it back (some sentences need revising).

Reviewer #3: The authors implement a purposive sampling of contact tracers, supervisors, and contacts of cases to document their experiences.

Overall, it’s well-written. I have some minor clarification concerns highlighted below.

Major concerns:

--I have some concerns about the generalizability of the results from the contacts.

--selection bias issues due to who was included and who wasn’t included.

On page 5, line 93, authors state “the first step is contact identification which involves asking the infected person….”

How? Interview? In person? By phone?

On page 5, line 95, “A contact can be anyone who has been in close proximity…”

What is a close contact definition?

On page 5, line 101, “In some cases, quarantine or isolation is required…”

So not all close contacts are required to isolate?

On page 5, in line 103, “Regular follow-up should be conducted with all contacts…”

Presumably these are close contacts? Not casual contacts?

On page 5, in line 104, “…for symptoms and test for signs of infection.”

Test close contacts or only those symptoms?

On page 6, line 122, “All contacts who were either asked to self-quarantine….”

Who is asked to self-quarantine? I get the impression it’s not everyone who is a close contact. This difference could be a source of selection bias.

What % of the close contacts were subsequently infected? What type of contact were they? Household? Neighbor? Work? This information would influence their impression of the importance of contact tracing.

On page 8, line 166: What % of tracers agreed to participate? How were they different from those who did not agree? 6 total non participants? For contacts, too?

In general, you really need a summary of the contact tracing process. This helps put things into context.

On page 9 table, adding a column showing the number who were contacted for participation

On page 12, only FIVE contacts included? And 4 of the 5 are older than all of the contact tracers and supervisors, just an observation. Not sure we can make any conclusions about contacts.

On page 14, line 258: screen everyone within 2km? Huh? Why would they all be considered contacts? Do the authors mean 2m?

On page 14, line 266: So every person within 2km of a case is tested and tested again after 14 days? And monitored daily? Even if this is 2m, not 2km, why only 5 contacts interviewed if this is the case?

6. PLOS authors have the option to publish the peer review history of their article (what does this mean?). If published, this will include your full peer review and any attached files.

Reviewer #1: **Yes: **Rafael Calvo

Reviewer #2: **Yes: **Diana Bright

Reviewer #3: No

---

## [Author Response · Author response to Decision Letter 0]

24 Dec 2020

We wish to thank the reviewers for their thorough review of our manuscripts. These comments are very useful and have helped in strengthening our manuscript. We have formatted the manuscript to conform with requirements of Plos One

Reviewers' comments:

Reviewer's Responses to Questions

Reviewer #1: This is a very nice piece of research about a very important and urgent topic.

In the rush to contain COVID, policy makers need to know what is actually happening in the ground. This paper contributes to that understanding.

The qualitative approach is appropriate as it provides the additional nuance needed to understand this complex multidimensional problem. The sample size is appropriate for this type of research, and the stakeholders interviewed seem to be a good selection.

Response: Thanks for the review

Comment: The writing is clear, although sometimes repetitive. The article could be made shorter. The major problem is that the themes arising from the study are not clear. The structure of the paper emphasizes aspects of the process. Is this what you found in the thematic analysis? Please explain how the data analysis led to the structure of reporting in the results section. This might need to be adapted to match the themes.

Response: We have provided enough details in data collection and analysis section to address these comments. These have been highlighted in each of the sections in this revised manuscript (see page 11 lines 222-231). We have also inserted a table to show the main and subthemes that emerged from the data (see Table 2) on page 12

Comment: It would also be interesting to identify what might the motivational factors that contribute to people quarantining according to the different stakeholders. These motivational drives are not made clear but would be very useful to policy makers.

Response: A section has been added on motivation for self-quarantine. Highlighted on page 20-22 of the revised manuscript. We have also added that to the discussion section of this revised manuscript.

.

Reviewer #2: 

This manuscript presents important experiences from people involved in contact tracing activities, from supervisors to contacts of confirmed COVID-19 cases. The topic is interesting given the unprecedented crisis the world is experiencing due the outbreak of a novel virus. It is important to share the experience of African countries as their 'community' approach to contact tracing has been different from other countries (maybe even more effective). There are very interesting points discussed in this paper but there are also some parts that could be improved by trying to describe in more detail the nuances of stakeholders' experiences, particularly those of the contact tracers and supervisors. For instance, was the training useful, if not, why? Did they follow all the guidance for contact tracing? If not, why not? Is there any data that can bring more insights into supervisors and contact tracers' experiences?

Also, there are some important points that were not fully discussed in the discussion section and could be addressed better.

Response: We have made some additions to the discussion section of this revised manuscript to address these comments. These have been highlighted under the sub-heading discussion.

Comment: Make sure to ask someone to proofread the manuscript before sending it back (some sentences need revising).

Response: We have done proofreading and copyediting 

Reviewer #3: 

Comment: The authors implement a purposive sampling of contact tracers, supervisors, and contacts of cases to document their experiences.

Overall, it’s well-written. I have some minor clarification concerns highlighted below.

Major concerns:

--I have some concerns about the generalizability of the results from the contacts.

--selection bias issues due to who was included and who wasn’t included.

Response: we have added a section on limitations of the study to address these comments. This is qualitative study which general limitation is generalizability of its findings (see the study limitations on page 30)

Comment: On page 5, line 93, authors state “the first step is contact identification which involves asking the infected person….”

How? Interview? In person? By phone?

Response: We have clarified this on page 5 line 97.

Comment: On page 5, line 95, “A contact can be anyone who has been in close proximity…”

What is a close contact definition?

Response: We have clarified this on page 5 line 98-99.

Comment: On page 5, line 101, “In some cases, quarantine or isolation is required…”

So not all close contacts are required to isolate?

Response: We have thrown more light to clarify that statement (see page 5 line 104-106)

Comment: On page 5, in line 103, “Regular follow-up should be conducted with all contacts…”

Presumably these are close contacts? Not casual contacts?

Response: This comment has also been addressed on page 5 line 107-108.

Comment: On page 5, in line 104, “…for symptoms and test for signs of infection.”

Test close contacts or only those symptoms?

Response: we have indicated that “Regular follow-up should be conducted with all the listed closed contacts to monitor for symptoms and test for signs of infection….” (see page 5 line 104-108)

Comment: On page 6, line 122, “All contacts who were either asked to self-quarantine….”

Who is asked to self-quarantine? I get the impression it’s not everyone who is a close contact. This difference could be a source of selection bias.

What % of the close contacts were subsequently infected? What type of contact were they? Household? Neighbor? Work? This information would influence their impression of the importance of contact tracing.

Response: As stated earlier, we addressed these issues per our responses to the comments on page 5. The focus of our study was how the contact tracing was done and experience of contact tracers and contact and not the number or percentages of contact who eventually tested positive for the disease (see Figure 1).

Comment: On page 8, line 166: What % of tracers agreed to participate? How were they different from those who did not agree? 6 total non-participants? For contacts, too?

Response: These comments have been addressed per our responses in page 10 line 209-211 (reference 16)

Comment: In general, you really need a summary of the contact tracing process. This helps put things into context.

Response: We have a flow chart to illustrate the contact tracing process (see Figure 1)

Comment: On page 9 table, adding a column showing the number who were contacted for participation

Response: Our response to your comments on page 209-211 (reference 16) addresses this. 

Comment: On page 12, only FIVE contacts included? And 4 of the 5 are older than all of the contact tracers and supervisors, just an observation. Not sure we can make any conclusions about contacts.

Response: Thanks for the observation. As indicted the data collection ended at the point of saturation. This acceptable practice in qualitative research (see reference 16)

Comment: On page 14, line 258: screen everyone within 2km? Huh? Why would they all be considered contacts? Do the authors mean 2m?

Response: This was a form of enhanced contact tracing. This was the practice in Ghana to ensure that the disease is really contained within any identified area that a case was confirmed. This was one of the innovations in the contact tracing in Ghana.

Comment: On page 14, line 266: So, every person within 2km of a case is tested and tested again after 14 days? And monitored daily? Even if this is 2m, not 2km, why only 5 contacts interviewed if this is the case?

Response: As stated earlier, interviews with contact ended at the point of saturation.

---

## [Editor Report · Decision Letter 1]

1 Feb 2021

Stakeholders perspective of, and experience with contact tracing for COVID-19 in Ghana: A qualitative study among contact tracers, supervisors, and contacts

PONE-D-20-20303R1

Dear Dr. Tabong,

We’re pleased to inform you that your manuscript has been judged scientifically suitable for publication and will be formally accepted for publication once it meets all outstanding technical requirements.

Kind regards,

Andrew Anglemyer, PhD

Guest Editor

PLOS ONE

---

## [Editor Report · Acceptance letter]

3 Feb 2021

PONE-D-20-20303R1 

Stakeholders perspective of, and experience with contact tracing for COVID-19 in Ghana: a qualitative study among contact tracers, supervisors, and contacts 

Dear Dr. Tabong:

I'm pleased to inform you that your manuscript has been deemed suitable for publication in PLOS ONE. Congratulations! Your manuscript is now with our production department. 

Kind regards, 

on behalf of

Dr Andrew Anglemyer 

Guest Editor

PLOS ONE